# CO_2_ Sorbents Based on Spherical Carbon and Photoactive Metal Oxides: Insight into Adsorption Capacity, Selectivity and Regenerability

**DOI:** 10.3390/molecules27206802

**Published:** 2022-10-11

**Authors:** Iwona Pełech, Ewelina Kusiak-Nejman, Piotr Staciwa, Daniel Sibera, Joanna Kapica-Kozar, Agnieszka Wanag, Filip Latzke, Karolina Pawłowska, Adrianna Michalska, Urszula Narkiewicz, Antoni W. Morawski

**Affiliations:** 1Department of Inorganic Chemical Technology and Environment Engineering, Faculty of Chemical Technology and Engineering, West Pomeranian University of Technology in Szczecin, Pułaskiego 10, 70-322 Szczecin, Poland; 2Faculty of Civil and Environmental Engineering, West Pomeranian University of Technology in Szczecin, al. Piastów 50a, 70-311 Szczecin, Poland

**Keywords:** carbon spheres, titanium dioxide, zinc oxide, physical adsorption, carbon dioxide, CO_2_/N_2_ selectivity

## Abstract

This work aimed to obtain hybrid composites based on photoactive metal oxide and carbon having adsorption properties. The materials, composed of titanium dioxide or zinc oxide and spherical carbon, were obtained from resorcinol-formaldehyde resin, treated in a solvothermal reactor heated with microwaves and then subjected to carbonization, were received. The functional groups of pure carbon spheres (unsaturated stretching C=C, stretching C−OH and C−H bending vibrations), CS/ZnO and CS/TiO_2_ samples were determined by FT-IR analysis. The characteristic bands for ZnO and TiO_2_ were observed below 1000 cm^−1^. The thermal oxidative properties are similar for TiO_2_- and ZnO-modified carbon spheres. We have observed that the increased carbon sphere content in nanocomposites results in starting the decomposition process at a lower temperature, therefore, nanocomposites have a broader combustion temperature range. The effect of the oxides’ addition to carbon spheres on their adsorption properties was evaluated in detail by examining CO_2_ adsorption from the gas phase. The selectivity of CO_2_ over N_2_ at a temperature of 25 °C and pressure of 1 bar (a novelty in testing CS-based sorbents) calculated for 3.00 CS/TiO_2_ and 4.00 CS/ZnO was 15.09 and 16.95, respectively. These nanocomposites exhibit excellent cyclic stability checked over 10 consecutive adsorption–desorption cycles.

## 1. Introduction

It is an undeniable fact, supported by the global scientific consensus, that increasing anthropogenic global greenhouse gas emissions leads to irreversible changes in the Earth’s climate. A limit has been set to prevent this: an increase in the global average temperature of less than 1.5 °C relative to pre-industrial times. For this to be possible, all sources of CO_2_ emissions must be decarbonized by 2050 [1]. The energy and heating sector is responsible for the largest share of global greenhouse gas emissions [2], most of which is carbon dioxide. Therefore, as long as fossil fuels remain the main energy source in the world, it will not be possible to achieve this goal. In order to reduce the concentration of CO_2_ in the atmosphere, it is necessary to switch to obtaining energy from renewable sources such as wind, solar or geothermal. However, nuclear power should be the basis due to one of the lowest carbon footprint values, averaging 66 g CO_2_/kWh, and its independence from weather conditions [3]. At the same time, it is necessary to introduce technologies to eliminate carbon dioxide from the exhaust gases of already existing emission sources.

Currently, research is being conducted, and technologies are being implemented in two directions: carbon capture and storage (CCS), and carbon capture and utilization (CCU) [4,5]. Separated CO_2_ can be stored in several ways. One of these is the mineralization of carbon dioxide into calcite, magnesite, dolomite or various forms of magnesium hydrocarbonates [6]. It can also be stored in depleted oil or gas fields and adsorbed into coal seams [7]. The second method involves the conversion of carbon dioxide into valuable products such as fuels, fertilizers or substrates for synthesis [8]. However, for CO_2_ to be stored or used, it must first be captured from the flue gas stream. Many technologies are dedicated to CO_2_ capture: absorption, adsorption, membrane separation, calcium looping, cryogenic and chemical looping [9,10]. However, despite intensive research and promising results, most of these technologies are not ready for industrial application in carbon dioxide capture or they are highly energy demanding.

The advantages of adsorption over other CO_2_ capture technologies are high adsorption capacity, regenerability, ease of operation of adsorption plants and material stability. Many materials have been tested as potential CO_2_ sorbents: MOFs, zeolites, metal oxides, mesoporous silicas or porous carbon materials [9]. Carbon-derived materials include nanotubes, nanowires, graphene and graphene oxides, and activated carbons or carbon spheres [11,12]. There are several parameters that a promising sorbent must fulfill: low manufacturing cost, high selectivity towards the adsorbed component, high sorption capacity, ease of regeneration and mechanical and thermal durability. Carbon spheres (CS) are materials that meet most of these criteria. They can be obtained in several ways: hydrothermally, arc discharge, chemical vapor deposition (CVD) or ultrasound treatment [13,14,15,16]. However, the most popular and most straightforward method to obtain them is the Stöber method, which was first used by Liu et al. [17]. In this process, the carbon source is a phenol-formaldehyde resin. Using this method, Wickaramantne and Jaroniec obtained carbon spheres with specific surface areas ranging from 390 to 669 m^2^/g (depending on the carbonization temperature). The sorption capacity of these materials was in the range of 1.4 to 3.02 mmol/g (from 3.6 × 10^−6^ to 4.5 × 10^−6^ mol/m^2^) [18]. To increase the efficiency of the adsorption process by carbon spheres, an activation process is necessary. This can be completed chemically or physically. Chemical activation can occur by treating the carbon spheres with compounds such as potassium hydroxide, potassium carbonate, potassium oxalate or zinc chloride [19,20,21,22]. Physical activation occurs by exposing the spheres to steam or carbon dioxide at the carbonization step [23,24]. Chemical activation significantly increases the specific surface area and the sorption capacity towards carbon dioxide at 0 and 25 °C. For example, in the work of Jaroniec et al., activation of spheres with potassium oxalate resulted in a specific surface area of 2130 m^2^/g, and carbon dioxide sorption capacity equals 6.6 mmol/g (3.09 × 10^−6^ mol/m^2^) [25]. The addition of a microwave excitation step made it possible to obtain spheres with an area of up to 1648 m^2^/g and CO_2_ sorption capacities ranging from 3.86 to 5.03 mmol/g at 25 °C (from 3.05 × 10^−6^ to 3.27 × 10^−6^ mol/m^2^) [26].

Carbon materials are known for their ability to improve the photocatalysis process, mainly due to the increased dispersion of the photocatalyst because of the high specific surface area of these materials. Moreover, their high electrical conductivity regulates the photocatalytic activity in the visible light range and attracts photogenerated electrons from the photocatalyst surface [27]. Therefore, carbon spheres modified with such photocatalysts as titanium dioxide or zinc oxide may be future materials capable of efficiently adsorbing and transforming CO_2_ into valuable compounds. In both cases, first attempts have already been made to obtain such composites. Zinc oxide-modified carbon spheres were first obtained in 2002 from an ion-exchangeable resin with the addition of a complex [Zn(NH_3_)_2_]^2+^. As a result, materials with a specific surface area ranging from 201 to 523 m^2^/g were obtained, in which a decrease in specific surface area with increasing ZnO content was noted [28]. ZnO/CS hybrid materials were obtained by Wang et al. in 2007 [29] using a surface-coating method. The low specific surface area of these materials (6–18 m^2^/g) resulted from the complete surface-coating of the carbon spheres with zinc oxide, forming various types of structures on them (e.g. described by the authors as “dandelions”). These composites showed high photocatalytic activity despite the low, specific surface area. In our previous studies, we investigated the effect of modification of carbon spheres with zinc oxide (obtained from ZnCl_2_ or Zn(NO_3_)_2_), and activation with potassium oxalate, on their surface parameters and the sorption capacity towards carbon dioxide. The specific surface area of the obtained materials ranged from 381 to 1233 m^2^/g, and the sorption capacity towards CO_2_ in the best case reached 2.69 mmol/g (2.18 × 10^−6^ mol/m^2^) at 40 °C [30,31]. For the modification of CS with titanium dioxide, Wang et al. [32] obtained spheres using a template hydrothermal method, which resulted in materials with a specific surface area of 386 m^2^/g, with 4.5 times higher photocatalytic activity than pure TiO_2_ (Degussa P25). The N-doped melamine-formaldehyde spheres supporting anatase TiO_2_, synthesized by a two-step solvothermal method obtained by Bi et al. [33], showed a surface area in the range of 70 to 106 m^2^/g, depending on the temperature of the hydrothermal treatment. Morawski et al. [34] obtained titanium dioxide-modified carbon spheres by microwave-assisted solvothermal synthesis using phenol-formaldehyde resin as the carbon source. The resulting material showed a sorption capacity towards CO_2_ of 3.94 mmol/g (9.12 × 10^−6^ mol/m^2^) (at 40 °C) and a specific surface area of 432 m^2^/g. For both the TiO_2_ and ZnO-modified CS, the presence of metal oxides on the carbon spheres’ surface reduces the composite materials’ total surface area and sorption capacity. However, this can be compensated for by the chemical activation process of the carbon spheres.

For this paper, we wanted to determine the role of the addition of selected metal oxides (ZnO and TiO_2_ in this particular case) in the adsorption removal of CO_2_ on modified carbon spheres. Additionally, to our knowledge, the CO_2_/N_2_ selectivity measurements performed and described in this paper are presented for the first time for spherical carbon materials modified with metal oxides. This study advances the general knowledge of CO_2_ physical adsorption utilizing carbon sphere-based composites.

## 2. Results and Discussion

The nanomaterials were subjected to FT-IR analysis to detect the various characteristic functional groups associated with the synthesized samples (spectra presented in Figure 1a,b). Both the CS/TiO_2_ and CS/ZnO groups of samples and reference samples (bare TiO_2_, ZnO and CS) are characterized by very similar FT-IR spectra. A broad band with a maximum at 3429 cm^−1^ is indexed to the bending and stretching modes of hydroxyl groups present on the surface of tested nanomaterials [35]. Considering that the bands with a maximum at 2924 and 2854 cm^−1^ are observed for all tested samples, they most likely correspond to the stretching vibration of hydroxyl compounds [36] related to the presence of KBr in a pellet. It is commonly known that KBr, due to its hygroscopic nature, can easily absorb water while preparing pellets. Other researchers also observed these peaks in KBr spectra [37]. However, in the case of materials with carbon spheres, these peaks can also be related to the C–H symmetric and asymmetric stretching vibrations, respectively, from the decomposition of ethyl alcohol used during the synthesis process to dissolve resorcinol, as described by Wang et al. [22] and Deshmukh et al. [38]. A similar observation was found for the peak at 1630 cm^−1^, which can be both assigned to the O–H bonding vibration proving the water adsorption [39] and the Zn–OH bending vibration in the case of ZnO samples [40], as well as the unsaturated C=C stretching vibrations, indicating that a carbonization process occurred during the formation of the spheres [41]. Peaks between 1360–1490 cm^−1^ correspond to the C−H bending vibrations [42]. The existing peaks at 960–1300 cm^−1^ are assigned to C−OH stretching and −OH bending vibrations [43]. The presence of −OH groups is related to many residual hydroxyl groups forming the hydrophilic surface. The characteristic peaks of TiO_2_ and ZnO are present below 1000 cm^−1^. The prominent peak of TiO_2_ is located in the 400–900 cm^−1^ range and corresponds to Ti−O stretching and Ti−O−Ti bridging stretching modes, while the characteristic band of ZnO is located in the range of 500 and 400 cm^−1^ and corresponds to the stretching mode of the Zn−O bond [44,45,46]. It is possible to observe different intensities of these peaks from the same group of samples. It does not mean that the amount of TiO_2_ or ZnO increases. The amount is the same for all composite nanomaterials, and this phenomenon is related to the CS amount—the peak intensity increases with decreases in the CS amount. With the higher amount of CS, the samples are darker and the measurement of the FT-IR spectra is more complicated.

The TG/DTG curves of the thermal oxidative stability of all the TiO_2_- or ZnO-modified samples with different carbon sphere contents are shown in Figure 2a,b and Figure 3a,b, respectively. In the case of reference TiO_2_ and ZnO semiconductors, only 2.30 and 1.25% weight losses were observed in a temperature range of 40–290 °C. That weight loss was attributed to the desorption of the physically adsorbed water [47]. For the unmodified carbon sphere sample (assigned as CS), the total weight loss of about 99.6% was observed between 369 and 730 °C, corresponding to the differential thermogravimetric (DTG) profile with a maximum at 674 °C, which could be assigned to breaking C–O bonds and desorption of residual organic compounds [48]. The results also show that both groups of samples have similar thermal behavior during the heat treatment, which means that TiO_2_ or ZnO photocatalysts are characterized by high thermal oxidative stability. For all samples, the first typical, however negligible, step of weight loss is observable between 100 and 220 °C, corresponding to the DTG profile with a maximum at 160 °C, which can be mainly ascribed to the desorption of moisture and other gases physically adsorbed on the surface of samples [49,50]. The next step occurred within the temperature range of 220 °C until 370 °C reaching the maximum at 270 °C (on DTG curves) due to the thermal decomposition of amorphous carbon [51].

It can be generally concluded that the thermal oxidative properties are broadly similar for TiO_2_- and ZnO-modified carbon spheres. The increased carbon sphere content in nanocomposites results in starting the decomposition process at a slightly lower temperature, and therefore, they have a broader combustion temperature range (from 410 to 650 °C for TiO_2_- and ZnO-modified carbon spheres with the lowest content carbon sphere and from 370 to 670 °C for TiO_2_- and ZnO-modified carbon spheres with the highest CS content). This shows that the addition of CS inhibits the pyrolysis of TiO_2_- and ZnO-modified composites. The total weight loss for the nanocomposites modified both with TiO_2_ or ZnO, containing the lowest carbon content of 0.50 CS, exhibits from 8 to 33% within the temperature range of 380–666 °C and 410–660 °C, respectively, corresponding to the DTG profiles centered at ca. 560 °C. With the increased carbon spheres’ content in nanocomposites, the total weight loss increased from 50 to 74 and 79%, and the maximum rate decomposition (DTG profile) shifted towards higher values, reaching the maximum at ca. 570 °C for the 1.20 CS/TiO_2_ and 1.60 CS/ZnO, and 580 °C for 3.00 CS/TiO_2_ and 4.00 CS/ZnO. This fact can be explained as follows: the higher the carbon and the lower the oxygen contents, the more oxygen during combustion is required, thus, the combustion was delayed, contributing to a higher DTG value.

Based on the low-temperature (−196 °C) nitrogen adsorption isotherms of the composites CS/TiO_2_ (Figure 4) and CS/ZnO (Figure 5), the values of the specific surface area (S_BET_), total pore volume (TPV), the volume of micropores (V_m_ < 2 nm), and volume of mesopores (V_meso_) were determined and listed in Table 1. For all the obtained samples together with the increase in carbon content, the calculated values of the specific surface area are also growing. The lowest values of S_BET_ were received for 0.05 CS/TiO_2_ and 0.05 CS/ZnO composites and amounted to 77 and 44 m^2^/g, respectively. The specific surface area of the composites based on titanium dioxide is slightly higher than that of composites based on zinc oxide. However, it was observed only for the materials with carbon content up to 1 g, probably due to the higher specific surface area of pure titanium dioxide. Above that value, the values of S_BET_ were similar for both types of composites. The highest S_BET_ were obtained for 3.00 CS/TiO_2_ and 4.00 CS/ZnO and amounted to 353 and 366 m^2^/g, respectively.

Significant differences for both series of materials were observed in the case of the total pore volume. For the samples based on titanium dioxide with the addition of 0.05 g of carbon spheres, TPV equaled 0.33 cm^3^/g. Similar values were also obtained for the composites with higher carbon content and, e.g. for the 1.20 CS/TiO_2_ material, the calculated value of TPV was 0.36 cm^3^/g. Contrary to the samples mentioned above, an increase in carbon content of up to 3 g resulted in a reduction in the total pore volume value, and the 3.00 CS/TiO_2_ TPV sample amounted only to 0.24 cm^3^. The samples based on zinc oxide and carbon spheres showed utterly different properties. The composites with carbon content ranging from 0.1 g to 1 g characterized the same TPV equaled 0.13 cm^3^/g, and these values were significantly lower than CS/TiO_2_ composites. Together with the increase in carbon content of up to 1.6 and 4.0 g, TPV values were higher and amounted to 0.18 and 0.21 cm^3^/g for 1.60 CS/ZnO and 4.00 CS/ZnO, respectively. The total pore volume for CS/TiO_2_ composites decreased, up to the value of 0.24 cm^3^/g (practically the same as for the pure carbon spheres), due to the low content of TiO_2_, characterized by V_meso_ equal to 0.38 cm^3^/g. Pure zinc oxide has far fewer mesopores, only 0.03 cm^3^/g. Therefore, lower TPV values were noticed compared to CS/TiO_2_ composites. For the sample 4.00 CS/ZnO, the total pore volume increased because the content of ZnO was low per 1 g of material, and TPV comes practically from the carbon.

According to IUPAC physisorption isotherm classification [52], CS/TiO_2_ composites, except the sample containing 3.00 g of carbon, exhibited a type II physisorption isotherm, which is characteristic for mesoporous or non-porous materials with lower N_2_ adsorption due to their much smaller BET surface area. In our case, the presence of carbon and titanium dioxide in the composites is responsible for the course of the isotherms. For pure TiO_2_, a significant volume of mesopores is characteristic. Then, it can be concluded that the shape of the isotherm is derived from the mesoporous character of the composites. In all the samples, H3-type hysteresis loops were observed, and their presence implies the occurrence of mesoporous or plate-like particles with slit-shaped pores [53,54]. Hysteresis loops disappear with the increased mesopores (as seen in Figure 4, inset). The isotherm obtained for 3.00 CS/TiO_2_ with the highest carbon contribution was of the mixed type I and II, indicating a high micropores’ content. The calculated value confirms these observations, and according to the data presented in Table 1, the volume of micropores with a diameter smaller than 2 nm increased together with the carbon content in the samples from 0.02 cm^3^/g to 0.16 cm^3^/g for the 0.05 CS/TiO_2_ and 3.00 CS/TiO_2_ sample. Simultaneously, the differences between the composites were noted in the case of the volume of mesopores with diameters from 2 to 50 nm. For the CS/TiO_2_ composites with the lowest carbon content, V_meso_ equaled 0.31 cm^3^/g, but for the sample with the addition of 1.2 g of carbon, it amounted to 0.25 cm^3^/g. A significant decrease in the mesopores content was recorded for the sample 3.00 CS/TiO_2_, for which the volume of macropores equaled only 0.08 cm^3^/g.

For the composites consisting of carbon spheres and zinc oxide, the obtained nitrogen isotherms were a mixture of type I and II, characteristic of micro- and macroporous materials [31]. In the samples with a low content of carbon spheres, we can observe a hysteresis loop type H3, which disappeared in the samples with a large amount of carbon spheres, from 1 to 4 g. For 1.00 CS/ZnO, 1.60 CS/ZnO, and 4.00 CS/ZnO samples, the nitrogen adsorption isotherms have the shape typical of microporous materials. According to the data presented in Table 1, the volume of micropores with a diameter smaller than 2 nm increased together with the carbon content in the samples, such as in the case of the CS/TiO_2_ samples, from 0.02 to 0.17 cm^3^/g for 0.10 CS/ZnO and 4.00 CS/ZnO sample, respectively. The opposite trend was observed in the case of the mesopores volume, such as in the case of the CS/TiO_2_ composites. However, the values were much lower and were in the range of 0.11 cm^3^/g for 0.10 CS/ZnO and 0.04 cm^3^/g for 4.00 CS/ZnO.

The adsorption properties of the obtained materials were evaluated in detail by examining the CO_2_ adsorption from the gas phase. The values of CO_2_ adsorption at 0 and 25 °C calculated based on the isotherms presented in Figure 6 (for CS/TiO_2_ composites) and Figure 7 (for CS/ZnO composites) are given in Table 1.

Due to the similar distribution of pores, the course of the isotherms obtained for the CS/TiO_2_ samples with the lowest carbon content (0.05 g and 0.1 g) was very similar. Increasing the carbon content up to 0.25 g resulted in the development of the surface area and mesoporosity of the composites, but the micropores’ content changed only slightly. As the contribution of carbon in the samples further increased, higher values of the adsorbed CO_2_ were detected. Similarly, the mesopores’ volume decreased for the 1.00 CS/TiO_2_ and 1.20 CS/TiO_2_ samples with a carbon content of 1 and 1.2 g (1.75 and 1.91 mmol/g of CO_2_ was adsorbed using these materials, respectively). The maximum amount of the adsorbed CO_2_ was reached for the sample with the lowest mesopores’ volume and equaled 2.45 mmol/g for the 3.00 CS/TiO_2_ material.

The cyclic adsorption–desorption behavior was carried out for samples modified with TiO_2_ or ZnO that achieved the highest CO_2_ sorption capacity at both 0 and 25 °C among all the tested materials in this study (3.00 CS/TiO_2_ and 4.00 CS/ZnO). The calculated CO_2_ capacities during ten consecutive adsorption–desorption cycles at 30 °C are shown in Figure 8.

During ten cycles, the adsorption capacity for both groups of tested samples was similarly constant, without noticeable degradation in the adsorption capacity of CO_2_. The average adsorption capacity after 10 consecutive runs reached 1.37 and 1.46 mmol/g for the 3.00 CS/TiO_2_ and 4.00 CS/ZnO, respectively. From these results, we can conclude that the carbon spheres exhibited significant CO_2_ adsorption capacity and showed excellent stability and regenerability—A crucial criterion for evaluating adsorbents for CO_2_ capture applications.

Although, as mentioned before, for the CS/TiO_2_ composites with a higher content of carbon the volume of the mesopores decreased, and for the 3.00 CS/TiO_2_ sample decreased drastically, the amount of the adsorbed CO_2_ increased. It clearly indicates that mesopores did not affect the amount of the adsorbed CO_2_, and the observed changes are related to the development of the microporosity of the composites. For this reason, the influence of the carbon concentration on the ultra-microporosity of the obtained composites’ pore size distributions was also studied. The values of the ultra-micropores’ volume were calculated based on the CO_2_ adsorption at 0 °C (Figure 6) using the NLDFT model, and the results are presented in Figure 9 and Table 1. The overall content of ultra-micropores for the samples with a lower carbon content (0.05 g; 0.1 g; 0.25 g) was similar and did not change significantly with the increased carbon content. Nevertheless, the contribution of 0.35 nm pores and 0.55 nm pores slightly increased. Considerable development of the ultra-microporosity was noticed when the carbon content reached 0.5 g and more. Higher contribution of carbon in the obtained composites caused the development of the pores 0.35 and 0.55 nm in size, reaching a maximum for the 3.00 CS/TiO_2_ sample. Hence, the adsorption properties of these samples towards CO_2_ have increased. Interestingly, the sample assigned as 1.20 CS/TiO_2_ was characterized by a very low content of 0.35 nm pores. However, the second peak starts at about 0.4 nm in size and indicates a broader range of the pores over 0.4 nm.

The same tendency can be noticed considering the CO_2_ adsorption on the CS/ZnO composites. It means that, with the increase in carbon content, CO_2_ adsorption values increased, reaching a maximum for the 4.00 CS/ZnO sample of 2.70 mmol/g. For this material, the lowest content of mesopores and the highest content of micropores were also observed. The pore size distributions of the CS/ZnO composites were also investigated and are given in Figure 10 and Table 1. Increasing the carbon content, a higher contribution of the 0.35 nm pores and 0.55 nm pores can be noticed in contrast to the 0.10 CS/ZnO sample. It should also be noted that comparing the two types of composites, for the CS/TiO_2_ samples with a carbon content up to 1 g, the values of CO_2_ adsorption obtained at 0 °C were slightly higher than for CS/ZnO. For example, using a 1.00 CS/TiO_2_ sample, 1.75 mmol/g of CO_2_ could be adsorbed at 0 °C, but using 1.00 CS ZnO material, only 1.54 mmol/g of CO_2_. Similar values were already noticed for the samples with carbon content above 1 g, regardless of the type of oxide present. Simultaneously for the CS/TiO_2_ material, a slightly higher content of micropores was noticed. It confirms a significant correlation between efficient CO_2_ adsorption at 0 °C and the presence of pores below 0.6 nm.

One important use of molecular models for adsorption is to examine multicomponent adsorption, as this type of adsorption is of obvious importance for practical applications. Adsorption isotherms, shown in Figure 11a,b were measured at a temperature of 25 °C up to a pressure of 1 bar.

Comparing the adsorption of CO_2_ and the adsorption of N_2_ for TiO_2_- and ZnO-modified CS at the same pressure and temperature conditions, we clearly see that the adsorption of CO_2_ (1.66 and 1.88 mmol/g, respectively) was much larger than that of N_2_ (0.28 mmol/g) across the entire pressure range, leading to the conclusion that the obtained nanocomposites exhibit good selectivity for CO_2_-over-N_2_ adsorption. Therefore, the S_IAST_ values calculated according to Equation (2), reached 15.90 and 16.95 for 3.00 CS/TiO_2_ and 4.00 CS/ZnO samples, respectively.

The CO_2_/N_2_ selectivity calculated for two representative samples (3.00 CS/TiO_2_ and 4.00 CS/ZnO) is shown in Figure 12. The highest CO_2_/N_2_ selectivity ratio at 0.1 bar was 49.5 and 61.6, and rapidly decreased with the pressure (up to ~0.3 bar), ultimately reaching values of 5.9 and 9.8 for 3.00 CS/TiO_2_ and 4.00 CS/ZnO, respectively, at the pressure around 1 bar. As expected, the selectivity ratio decreased due to the small gas molecules being well absorbed in narrow pores at low pressures. Moreover, CO_2_ has a stronger tendency to combine with carbon pores than N_2_. Once the adsorption sites are unavailable with pressure increase, the remaining gas molecules are squeezed into the confined pores [55].

## 3. Materials and Methods

### 3.1. Materials

Carbon spheres (CS) were obtained from resorcinol and formaldehyde. For this purpose, 2.4 g of resorcinol was dissolved in an aqueous alcohol solution composed of 240 mL distilled water and 96 mL of ethanol. To adjust pH to ~9, ammonium hydroxide (25 wt.%) was slowly dropped into the beaker. Then, 3.6 mL of formaldehyde (37 wt.%) was added, and the whole was mixed using a magnetic stirrer at ambient conditions to facilitate a polycondensation reaction. After 24 h, the mixture was transferred into a microwave reactor (Ertec-Poland, Wrocław, Poland), and the treatment there was carried out for 15 min under a reaction pressure of 20 atm. Next, the products were dried for 24 h at 80 °C and then carbonized in a HST 12/400 Carbolite high-temperature furnace (Carbolite Gero Ltd., Sheffield, UK) under argon atmosphere with the temperature increasing from 20 to 350 °C at a heating rate of 1 °C/min and holding time of 2 h and from 350 to 700 °C at a heating rate of 1 °C/min. After 2 h, the sample was cooled to room temperature under an argon atmosphere. The as-obtained material was washed with distilled water and dried for 48 h at 80 °C in air. The as-prepared carbon spheres in the appropriate amount (from 0.05 to 4 g) were placed into the beaker with 100 mL of ethylene alcohol and stirred for 30 min. Then, 1 g of titanium dioxide (AEROXIDE^®^ TiO_2_ P25; Evonik Industries AG, Essen, Germany) or zinc oxide nanopowder (<100 nm particle size; Sigma-Aldrich, St. Louis, MO, USA) was added, and the whole was stirred again, but this time for 24 h. Afterwards, the obtained mixture was transferred into a microwave-assisted solvothermal reactor (Ertec, Wrocław, Poland), and the treatment there was carried out for 15 min under a reaction pressure of 20 atm. Finally, the obtained composites were dried for 48 h at 80 °C in air.

### 3.2. Characterization Methods

The chemical structure of tested materials was investigated by the Thermo Scientific Nicolet 380 spectrometer (Thermo Fisher Scientific Inc., Waltham, MA, USA). Samples were prepared by grinding the investigated material with KBr (Merck KGaA, Darmstadt, Germany) and forming pellets. The procedure of pellets’ preparation was described in detail in our previous work [30]. For better visualization, all spectra were multiplied by 5.

Thermogravimetric analysis (TG/DTG) was conducted using NETZSCH STA 449 F3 Jupiter (Erich NETZSCH GmbH & Co. Holding KG, Selb, Germany). For this purpose, the sample mass of about 10 mg was placed in an open corundum crucible with a corresponding empty referent pan. Thermal oxidative stability of all samples was measured at a heating rate of 10 °C/min under a flow of air atmosphere (70 mL/min). The analyses were performed over the temperature range of 30–1000 °C.

Surface properties were determined using N_2_ adsorption/desorption isotherms performed on a QUADRASORB evoTM Gas Sorption automatic system (Quantachrome Instruments, Boynton Beach, FL, USA) at −196 °C. Before each adsorption experiment, samples were outgassed at 250 °C under a vacuum of 1 × 10^−5^ mbar for 12 h using a MasterPrep multi-zone flow/vacuum degasser from Quantachrome Instruments to remove adsorbed species that could intervene in the adsorption processes. The surface area (S_BET_) was determined in the relative pressure range of 0.05–0.3 and calculated based on Brunauer–Emmett–Teller (BET) equation. The total pore volume (TPV) was calculated from the volume of nitrogen held at the highest relative pressure (*p*/*p*_0_ = 0.99). The volume of micropores V_m_ with diameter below 2 nm was calculated as a result of integrating the pore volume distribution function using the DFT method; the volume of mesopores V_meso_ with diameter from 2 to 50 nm was calculated from the difference in the total pore volume TPV and the volume of micropores V_m_.

Carbon dioxide adsorption isotherms at 0 and 25 °C were measured using the same Quadrasorb™ automatic system (Quantachrome Instruments, Boynton Beach, FL, USA) mentioned above in the pressure range between 0.01 and 0.98 bar. From CO_2_ adsorption, the isotherms at 0 °C, pore size distribution (PSD), and the volume of ultra-micropores V_s_ below 1.0 nm (<1 nm) were determined and calculated by integrating the pore volume distribution function using the NLDFT method.

The carbon dioxide cyclic adsorption–desorption measurements were investigated using a thermogravimetric analyzer NETZSCH STA 449 F3 Jupiter. First, the tested samples were pre-dried at 105 °C in pure argon flow (70 mL/min) for 60 min to remove physiosorbed moisture and/or CO_2_ (from the surface and/or pores), then cooled to the adsorption temperature (30 °C). After the temperature stabilized, the argon flow was reduced to 10 mL/min (used as a protective gas), switched on pure CO_2_ and held for 60 min in a flow of 90 mL/min. After adsorption, the gas was switched from CO_2_ to argon (70 mL/min), and the temperature increased to 105 °C to desorb the CO_2_. The 10 consecutive adsorption–desorption cycles were performed to evaluate the stability of tested adsorbents, and after each adsorption–desorption run, the sample weight was recorded to calculate the CO_2_ uptake.

Additionally, to test the selectivity of CO_2_ adsorption over N_2_ carried out the nitrogen adsorption–desorption measurements at 25 °C up to a pressure of 1 bar, using the Quantachrome Instruments Quadrasorb apparatus, were performed for 3.00 CS/TiO_2_ and 4.00 CS/ZnO samples with the best CO_2_ adsorption at 0 and 25 °C. To determine the selectivity of the CO_2_ and N_2_ molecules, we compared the CO_2_ and N_2_ adsorption from the binary mixture simulations with the CO_2_ and N_2_ adsorption from the single-component isotherms.

The ideal adsorption solution theory (IAST) was used for the prediction of CO_2_/N_2_ binary mixture adsorption equilibrium using single components adsorption data according to (Equation (1)):(1)$$S_{CO_{2}}=\frac{q_{CO_{2}\left(p\right)}}{q_{N_{2}\left(p\right)}}$$
where: *q_i_*(*p*)—the adsorption capacity (mmol/g) at the same partial pressure *p*.

Developed by Myers and Prausnitz [56], the ideal adsorbed solution theory (IAST) is widely used for readily predicting multicomponent adsorption isotherms from only pure-component adsorption isotherms at the same temperature. Carbon dioxide selectivity for a gas mixture containing 85% of N_2_ and 15% of CO_2_ from 0 to 1 bar was calculated according to Equation (2):(2)$$S_{0.15:0.85}=\frac{q_{CO_{2}at0.15\mathrm{bar}}}{q_{N_{2}at0.85\mathrm{bar}}}\times \frac{0.85}{0.15}$$
where: $q_{CO_{2}at0.15\mathrm{bar}}$—the CO_2_ uptake (mmol/g) at the partial pressure of 0.15 bar, $q_{N_{2}at0.85\mathrm{bar}}$—the N_2_ uptake (mmol/g) at the partial pressure of 0.85 bar.

During the measurements, air, argon, carbon dioxide, and nitrogen gases with a purity of 99.995% (Air Liquide, Białystok-Zaścianki, Poland) were used.

## 4. Conclusions

The properties of the composites based on carbon spheres and titanium dioxide, or carbon spheres and zinc oxide were compared. The thermal oxidative analysis and FT/IR measurements confirmed the presence of different carbon functional groups in tested samples, mainly unsaturated C=C from amorphous carbon and C−O(H). FT/IR analysis revealed the presence of other groups in the form of C−H. Additionally, the increased spherical carbon content in composites further broadens the combustion temperature range (the oxidative decomposition process starts at a lower temperature). However, the more metal oxide added, the narrower the combustion range. Notably, the detailed structural studies showed that the much higher values of total pore volume for the materials with the addition of titanium dioxide, caused by the higher contribution of mesopores in the samples, did not influence the carbon dioxide adsorption. Simultaneously, in the case of CS/TiO_2_ material with the highest carbon content, for which a significant decrease in the volume of mesopores was observed, the decrease in CO_2_ adsorption was not noticed. Generally, slightly lower values of CO_2_ adsorption were noted for CS/ZnO samples with carbon content up to 1 g than for CS/TiO_2_ due to a slightly lower content of micropores for these materials. For the composites with a carbon content above 1 g, the same amount of CO_2_ was adsorbed regardless of the type of oxide present in the composite. In both groups of composites, the incorporation of higher carbon content led to the development of ultra-microporosity. It was found that the size of micropores played an essential role, and the pores below 0.6 nm were responsible for efficient CO_2_ adsorption. Using the representative 3.00 CS/TiO_2_ and 4.00 CS/ZnO samples with the highest adsorption capacity at 0 and 25 °C, it was also found that the synthesized composite materials were characterized by very good stability and regenerability (measured during ten consecutive CO_2_ adsorption–desorption cycles). The CO_2_/N_2_ selectivity checked for the first time for spherical carbon-based composites showed a stronger tendency of CO_2_ to combine with carbon pores than N_2_.

## Figures and Tables

**Figure 1 molecules-27-06802-f001:**
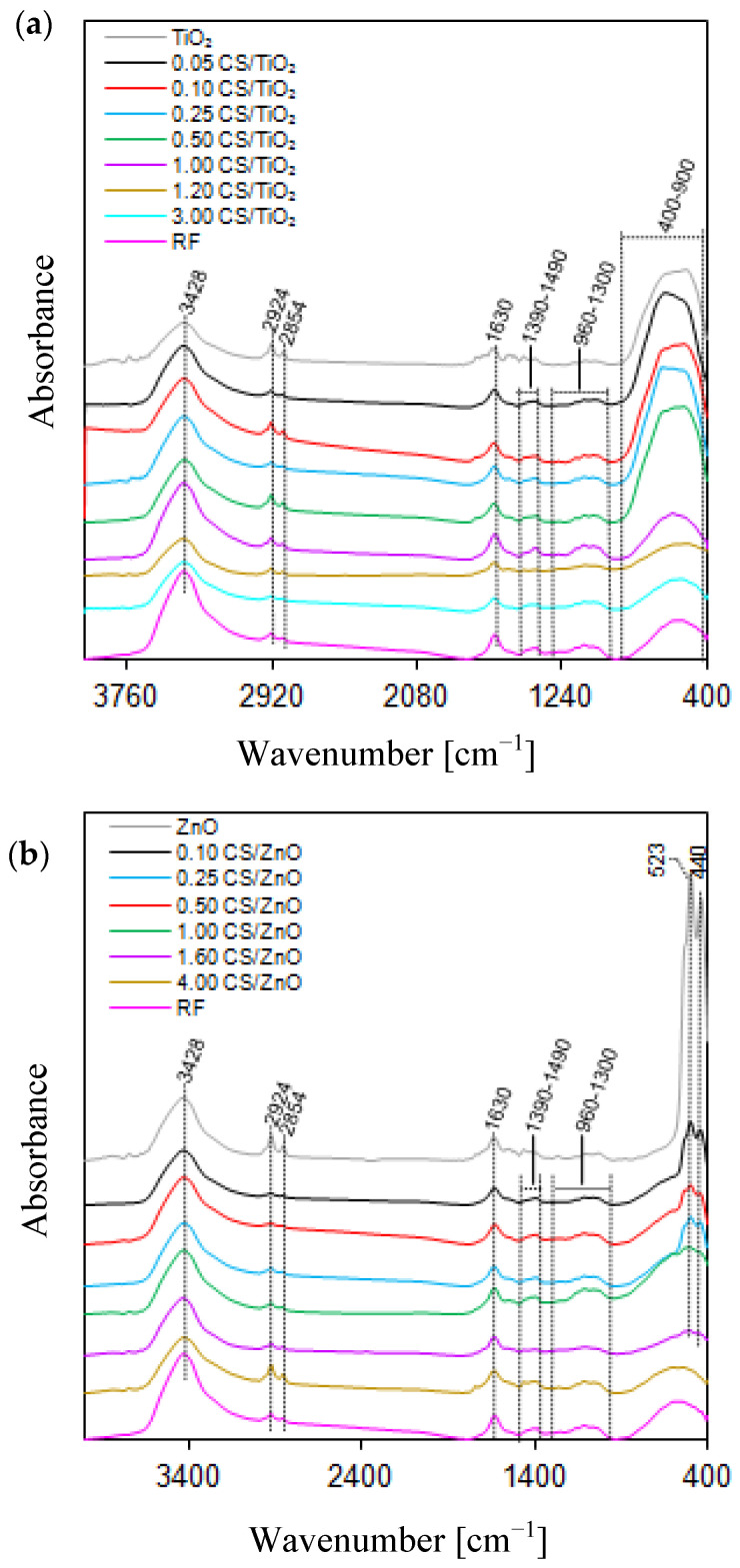
FT-IR spectrum of (**a**) CS/TiO_2_ and (**b**) CS/ZnO samples.

**Figure 2 molecules-27-06802-f002:**
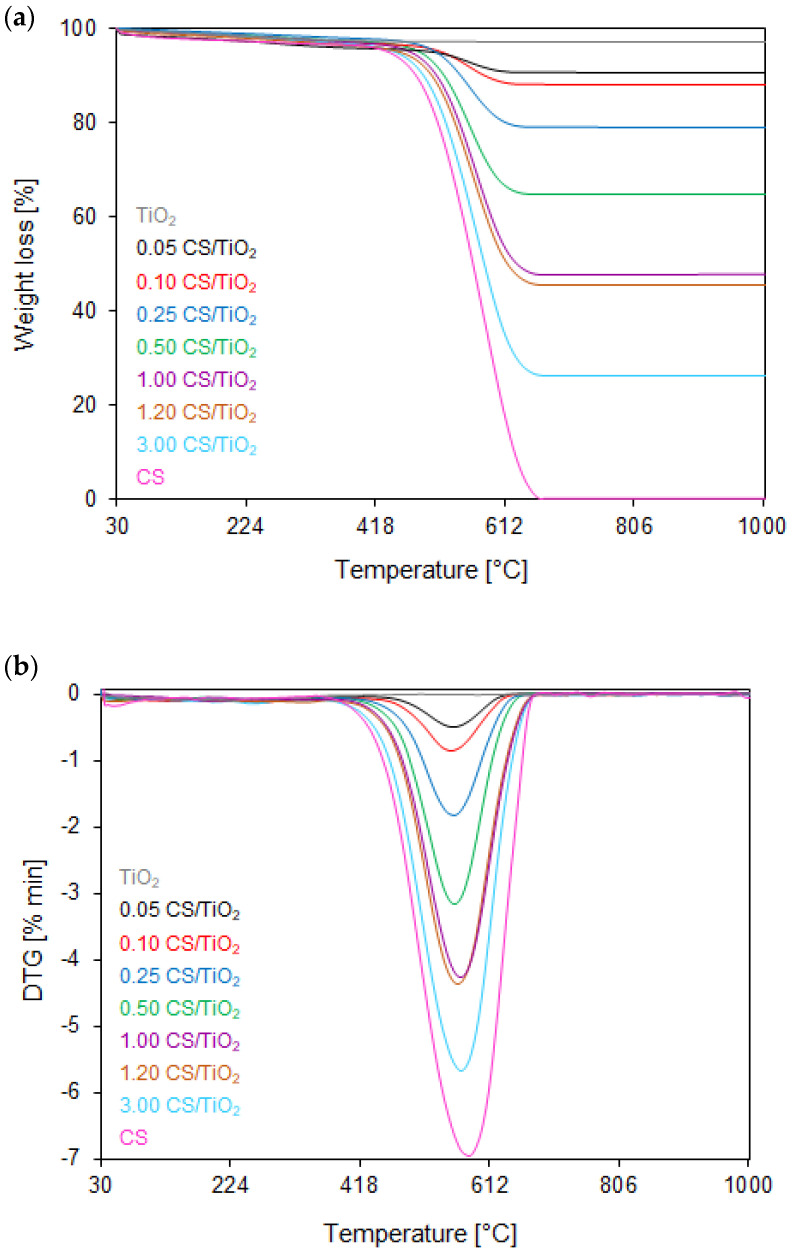
(**a**) TG and (**b**) DTG curves of reference TiO_2_, unmodified CS, and TiO_2_-modified nanocomposites with different carbon sphere contents measured in an oxidizing atmosphere.

**Figure 3 molecules-27-06802-f003:**
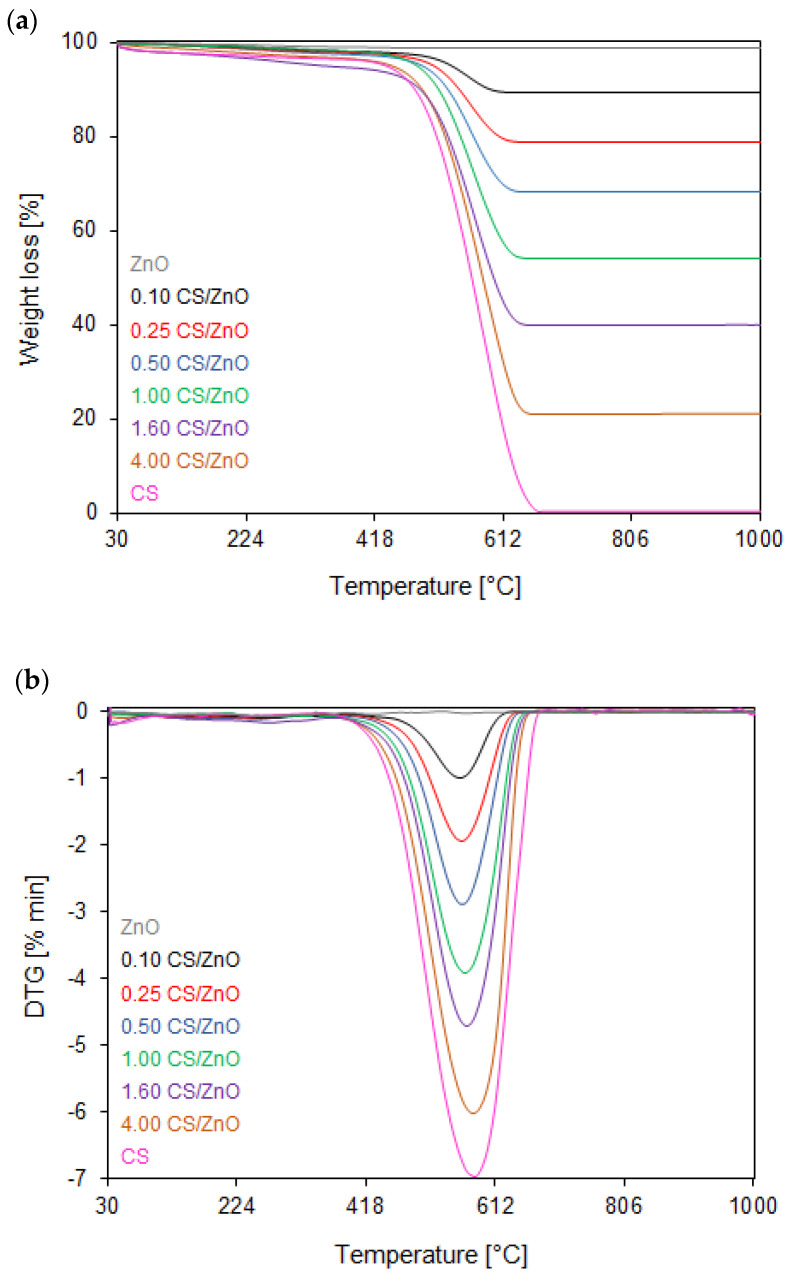
(**a**) TG and (**b**) DTG curves of reference ZnO, unmodified CS, and ZnO-modified nanocomposites with different carbon sphere contents measured in an oxidizing atmosphere.

**Figure 4 molecules-27-06802-f004:**
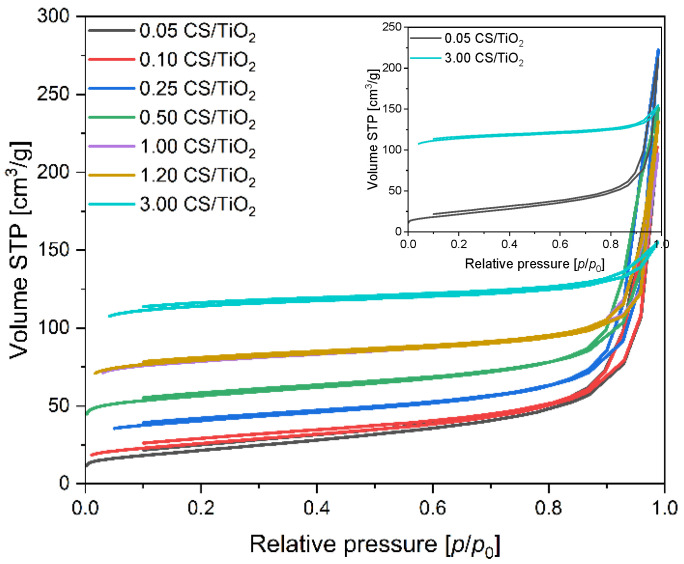
Nitrogen adsorption–desorption isotherms of all tested CS/TiO_2_ composites, and selected 0.05 and 3.00 CS/TiO_2_ samples (inset).

**Figure 5 molecules-27-06802-f005:**
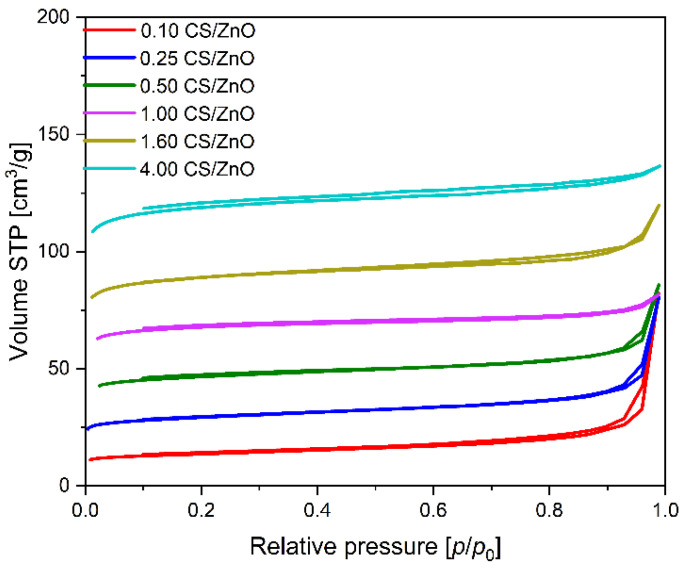
Nitrogen adsorption–desorption isotherms of CS/ZnO composites.

**Figure 6 molecules-27-06802-f006:**
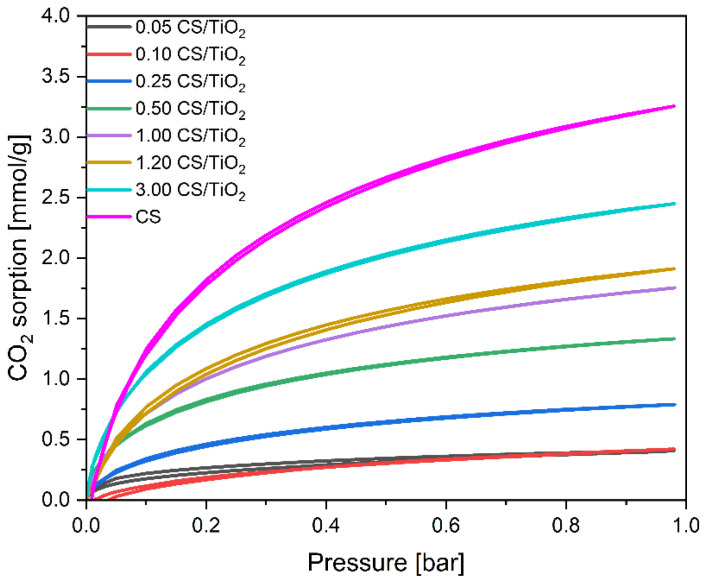
CO_2_ sorption isotherms of the unmodified CS and obtained CS/TiO_2_ composites.

**Figure 7 molecules-27-06802-f007:**
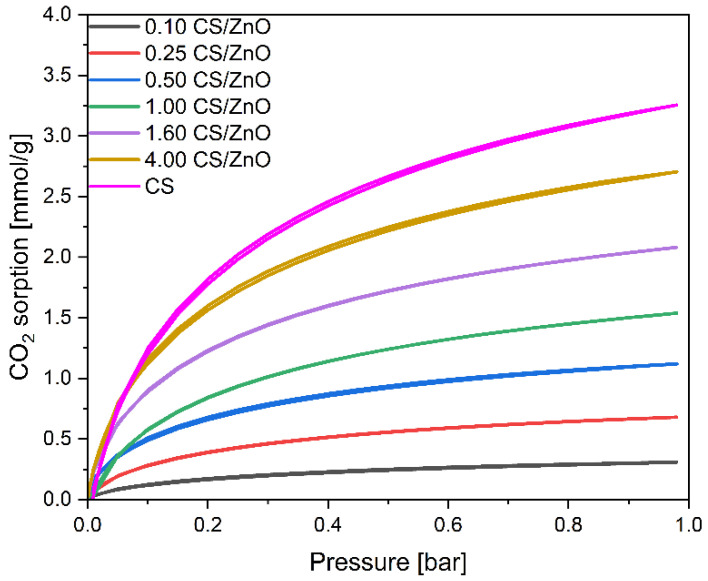
CO_2_ sorption isotherms of the reference CS and the obtained CS/ZnO composites.

**Figure 8 molecules-27-06802-f008:**
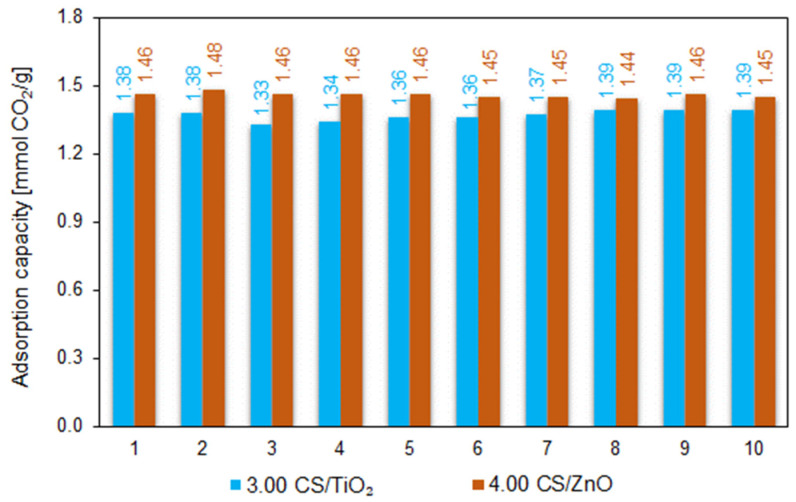
Multicyclic stability of CO_2_ adsorption–desorption measured at 30 °C for 4.00 CS/TiO_2_ and 3.00 CS/TiO_2_ samples.

**Figure 9 molecules-27-06802-f009:**
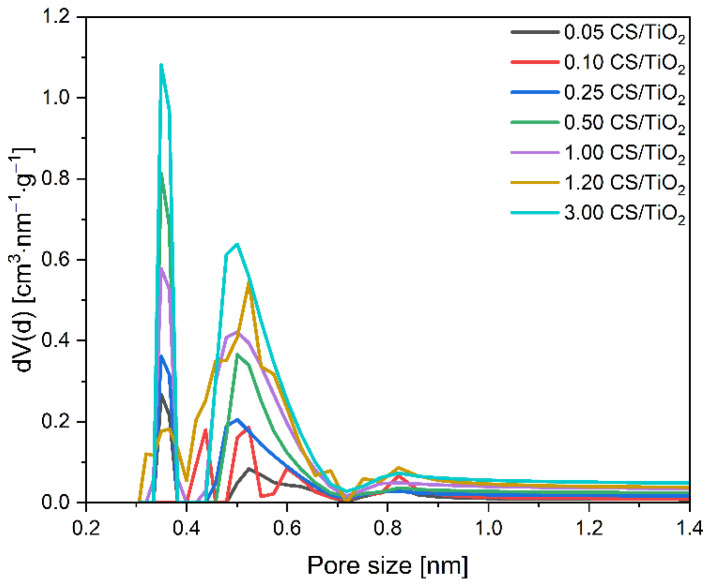
Pore size distributions of the obtained CS/TiO_2_ composites.

**Figure 10 molecules-27-06802-f010:**
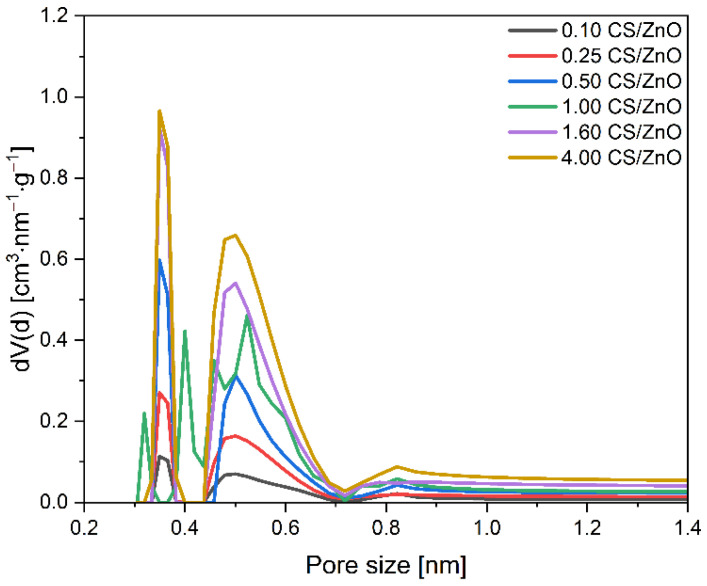
Pore size distributions of the obtained CS/ZnO composites.

**Figure 11 molecules-27-06802-f011:**
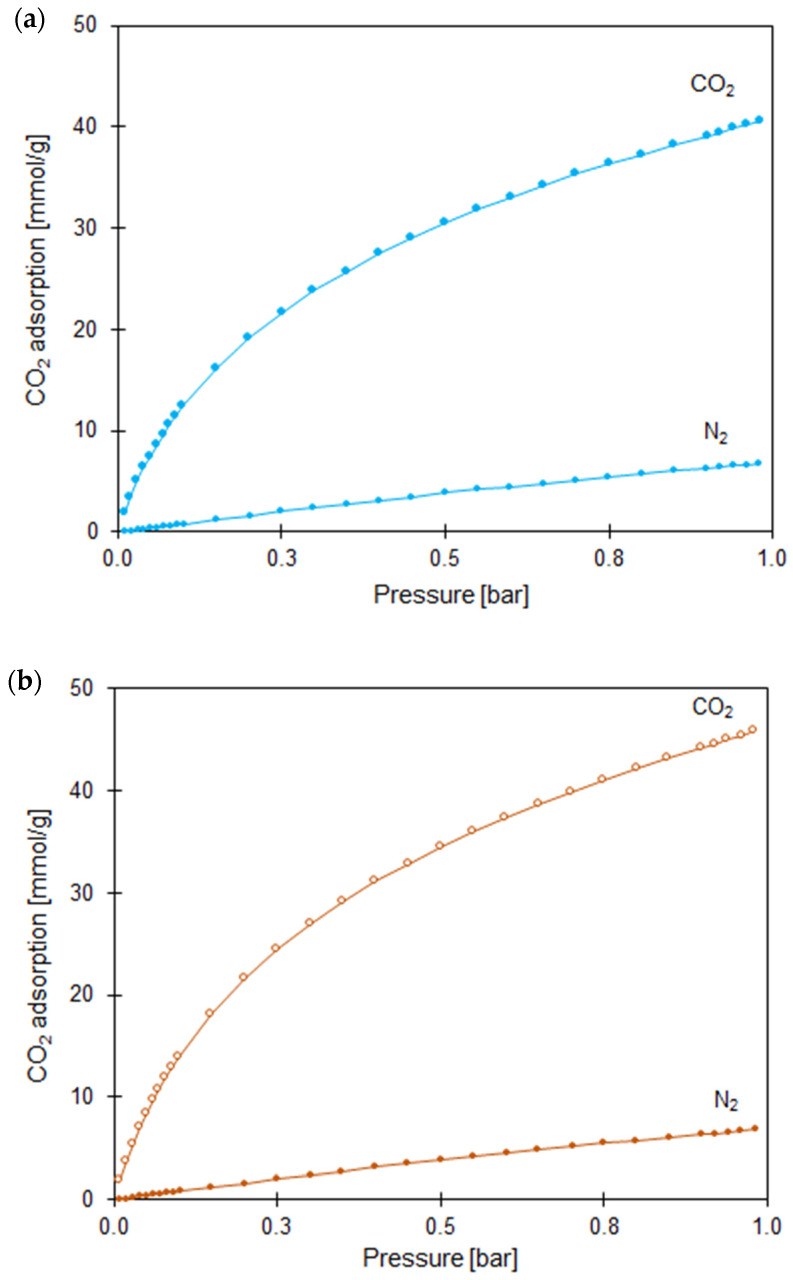
A comparison of single component CO_2_ and N_2_ adsorption isotherms for (**a**) 3.00 CS/TiO_2_ and (**b**) 4.00 CS/ZnO samples.

**Figure 12 molecules-27-06802-f012:**
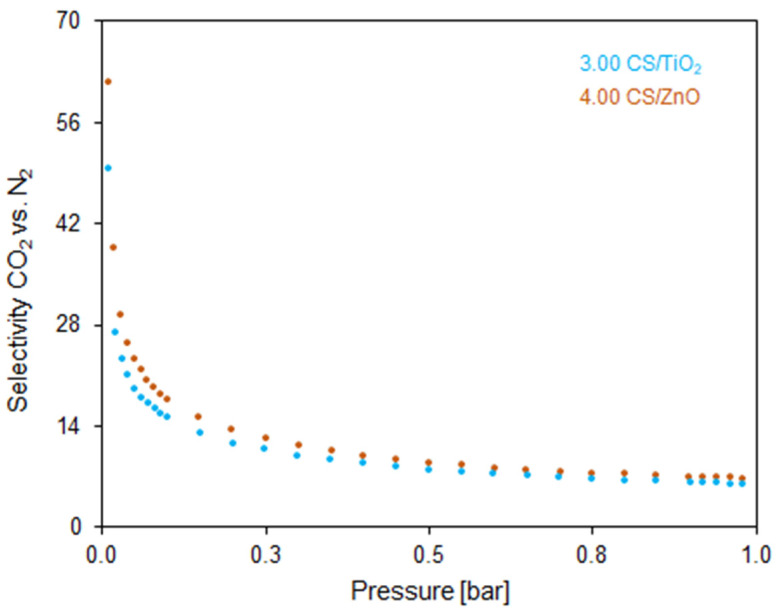
Selectivity of CO_2_ over N_2_ at 25 °C for 3.00 CS/TiO_2_ and 4.00 CS/TiO_2_ samples.

**Table 1 molecules-27-06802-t001:** Textural parameters and CO_2_ sorption capacities for the activated carbon spheres.

	StructuralProperties →	S_BET_	TPV	V_s_ (<1 nm)	V_m_ (<2 nm)	V_meso_	CO_2_ 0 °C	CO_2_ 25 °C
Sample Code ↓		(m^2^/g)	(cm^3^/g)	(cm^3^/g)	(cm^3^/g)	(cm^3^/g)	(mmol/g)	(mmol/g)
CS		455	0.26	0.19	0.22	0.04	3.25	2.43
TiO_2_		54	0.40	-	0.02	0.38	-	-
ZnO		11	0.03	-	0.00	0.03	-	-
0.05 CS/TiO_2_		77	0.33	0.02	0.02	0.31	0.41	0.29
0.10 CS/TiO_2_		89	0.34	0.03	0.03	0.31	0.42	0.33
0.25 CS/TiO_2_		132	0.43	0.04	0.04	0.39	0.79	0.54
0.50 CS/TiO_2_		181	0.37	0.07	0.07	0.30	1.33	0.90
1.00 CS/TiO_2_		247	0.33	0.10	0.11	0.22	1.75	1.22
1.20 CS/TiO_2_		248	0.36	0.11	0.11	0.25	1.91	1.36
3.00 CS/TiO_2_		353	0.24	0.13	0.16	0.08	2.45	1.66
0.10 CS/ZnO		44	0.13	0.02	0.02	0.11	0.31	0.21
0.25 CS/ZnO		92	0.13	0.04	0.04	0.09	0.68	0.46
0.50 CS/ZnO		145	0.13	0.06	0.06	0.07	1.12	0.74
1.00 CS/ZnO		209	0.13	0.09	0.10	0.03	1.54	1.11
1.60 CS/ZnO		275	0.18	0.11	0.12	0.06	2.08	1.41
4.00 CS/ZnO		366	0.21	0.15	0.17	0.04	2.70	1.88

S_BET_—specific surface area; TPV—total pore volume; Vs—the volume of ultra-micropores with a diameter smaller than 1 nm; V_m_—the volume of micropores with a diameter smaller than 2 nm; V_meso_—the volume of mesopores with a diameter from 2 to 50 nm.

## Data Availability

The data presented in this study will be available upon request.

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
