# Peer review of "CO_2_ Sorbents Based on Spherical Carbon and Photoactive Metal Oxides: Insight into Adsorption Capacity, Selectivity and Regenerability"

_molecules, 2022, doi:10.3390/molecules27206802_

Round 1
Reviewer 1 Report
In this manuscript, authors reported synthesis of CS/TiO2 and CS/ZnO composites and investigated their performance for CO2 adsorption from the gas phase. Before accepting it to be published, some major revisions are needed. Detailed comments are listed as follow:
1. The image resolutions of all figures should be greatly improved.
2. Why absorptions of C-H bonds appear at 2854 and 2924 cm-1 in the pure TiO2 and ZnO?
3. The CO2 sorption isotherms of pure CS, TiO2 and ZnO should be added into Fig. 6 and Fig. 7.
4. As shown in Table 1, since the CS has the highest CO2 adsorption capacity, why they added metal oxides into CS?
Author Response
Dear Reviewer 1,
please find the attached file with responses to your questions and suggestions.

Reviewer 2 Report
The current manuscript by Pelech et al. represents a short study on synthesizing activated carbon particles via ZnO and TiO2. The procedures for the synthesis and sorption measurements are clearly presented. The authors achieve high selectivity of their material with respect to N2/CO2 sorption at ambient conditions for 10 cycles. I recommend the manuscript for publication after the following minor issues are addressed:
(1) Please compare your materials in terms of efficiency to other authors – this is partially done in the introduction, but it is not clear how the efficiency of the materials scale to other authors (mole CO2 per m2 for instance is a better measure of efficiency than mol/g, since it includes the specific area)
(2) It is not clear how the authors prove the spherical shape of the particles – please clarify this statement and illustrate it appropriately
(3) The references citations should be corrected – missing brackets, missing citations, etc.
(4) Please correct misuse of different fonts along text due to copy-paste from different sections or documents.
Author Response
Dear Reviewer 2,
Responses to your comments and suggestions are included in the attached file.
Yours faithfully,
Ewelina Kusiak-Nejman, corresponding author

Reviewer 3 Report
-
CHeck ref in lines 63-74
-
A schematic explaining the adsorption/desorption of CO2 in the two modified CS could help make it clearer to the reader why the adsorption capacities of modified CS is different from CS on their own
-
Adding a reference for pure CS might be useful to the reader to understand the effect of the metal oxides better.
Author Response
Dear Reviewer 3,
In the attachment, you will find responses to the questions and suggestions in the review (marked in red).
Yours faithfully,
Ewelina Kusiak-Nejman, corresponding author

Round 2
Reviewer 1 Report
They have answered the questions.